# Oncological Outcomes of Robotic Lobectomy and Radical Lymphadenectomy for Early-Stage Non-Small Cell Lung Cancer

**DOI:** 10.3390/jcm11082173

**Published:** 2022-04-13

**Authors:** Filippo Tommaso Gallina, Riccardo Tajè, Daniele Forcella, Felicita Corzani, Virna Cerasoli, Paolo Visca, Cecilia Coccia, Federico Pierconti, Isabella Sperduti, Fabiana Letizia Cecere, Federico Cappuzzo, Enrico Melis, Francesco Facciolo

**Affiliations:** 1Thoracic Surgery Unit, IRCCS Regina Elena National Cancer Institute, 00144 Rome, Italy; r.taje@virgilio.it (R.T.); daniele.forcella@ifo.it (D.F.); felicita.corzani@ifo.it (F.C.); virna.cerasoli@ifo.it (V.C.); enrico.melis@ifo.it (E.M.); francesco.facciolo@ifo.it (F.F.); 2Department of Pathology, IRCCS Regina Elena National Cancer Institute, 00144 Rome, Italy; paolo.visca@ifo.it; 3Anesthesiology and Intensive Care Unit, IRCCS Regina Elena National Cancer Institute, 00144 Rome, Italy; cecilia.coccia@ifo.it (C.C.); federico.pierconti@ifo.it (F.P.); 4Biostatistics, IRCCS Regina Elena National Cancer Institute, 00144 Rome, Italy; isabella.sperduti@ifo.it; 5Medical Oncology 1, IRCCS Regina Elena National Cancer Institute, 00144 Rome, Italy; fabiana.cecere@ifo.it; 6Medical Oncology 2, IRCCS Regina Elena National Cancer Institute, 00144 Rome, Italy; federico.cappuzzo@ifo.it

**Keywords:** early-stage NSCLC, robotic surgery, nodal upstaging, lobectomy

## Abstract

Background: While the thoracotomy approach was considered the gold standard until two decades ago, robotic surgery has increasingly strengthened its role in lung cancer treatment, improving patients’ peri-operative outcomes. In this study, we report our experience in robotic lobectomy for early-stage non-small cell lung cancer, with particular attention to oncological outcomes and nodal upstaging rate. Methods: We retrospectively reviewed patients who underwent lobectomy and radical lymphadenectomy at our Institute between 2016 and 2020. We selected 299 patients who met the inclusion criteria of the study. We analyzed the demographic features of the groups as well as their nodal upstaging rate after pathological examination. Then, we analyzed disease-free and overall survival of the entire enrolled patient population and we compared the same oncological outcomes in the upstaging and the non-upstaging group. Results: A total of 299 patients who underwent robotic lobectomy were enrolled. After surgery, 55 patients reported nodal hilar or mediastinal upstaging. The 3-year overall survival of the entire population was 82.8%. The upstaging group and the non-upstaging group were homogeneous for age, gender, smoking habits, clinical stage, tumor site, tumor histology. The non-upstaging group had better OS (*p* = 0.004) and DFS (*p* < 0.0001). Conclusion: Our results show that robotic surgery is a safe and feasible approach for the treatment of early-stage NSCLC, especially for its accuracy in mediastinal lymphadenectomy. The oncological outcomes were encouraging and consistent with previous findings.

## 1. Introduction

Lung cancer is the leading cause of cancer-related deaths in the world. Non-Small Cell Lung Cancer (NSCLC) represents 80% of all lung cancers. Despite the progress in therapies, patients with NSCLC still present an estimated 5-year overall survival rate less than 25% when considering all tumor stages. Regarding stage I and stage II, the estimated 5-year overall survival rate reported in the literature is around 80% [1]. The radical treatment of early-stage NSCLC consists in lobectomy and hilum–mediastinal lymphadenectomy [2].

In the last two decades, minimally invasive surgery in the treatment of early-stage NSCLC has become increasingly common. Video-Assisted Thoracic Surgery (VATS) lobectomy for the treatment of early-stage NSCLC was introduced in the early 1990s. Since then, a variety of VATS procedures have been performed including tri-, bi-, uniportal, and, more recently, subxiphoid approaches [3]. All these procedures allow a shorter hospital stay, lower morbidity, decreased postoperative pain, and better cosmesis. Controversies still remain regarding oncologic radicality when compared to thoracotomy [4].

Since its introduction, robot-assisted thoracic surgery has rapidly redesigned the concept of minimally invasive thoracic surgery [5]. A better anatomical visualization, thanks to three-dimensional technology, a wider maneuverability range of the instruments, and more precise movements are only the most known advantages of robotic technology. Moreover, robot-assisted thoracic surgery has been associated with a comprehensively shorter length of hospital stay, a decreased number of post-operative complications, and decreased mortality when compared with open techniques, while differences with thoracoscopic approaches are still under investigation [6]. In this respect, the latest studies that compared the perioperative outcomes of robotic thoracic surgery with those of thoracoscopy showed less blood loss, a lower conversion rate, a lower overall complication rate, and an improved lymph node harvest in the robotic arm, though with a higher overall costs [7,8].

Since the beginning of the robot-assisted thoracic surgery program in 2016 at our institution, almost 800 robotic thoracic procedures have been accomplished. Hereby, we review our experience in robotic lobectomy and radical lymphadenectomy for early-stage NSCLC.

## 2. Materials and Methods

From January 2016 to December 2021, a total of 557 patients underwent robotic lobectomy for suspected non-small cell lung cancer; 276 patients were male, and 281 were female. Their median age was 66.9 ± 9.5. The aim of this study was to evaluate the oncological outcomes of patients undergoing pulmonary lobectomy and lymphadenectomy for early-stage lung cancer at our institution. Particularly, radical lymphadenectomy characteristics such as the number of dissected nodal stations and harvested lymph nodes, as well as nodal upstaging were analyzed. The clinical and surgical characteristics of the patients with and without pathologic evidence of nodal upstaging (upstaging and non-upstaging groups) were compared. Finally, the 3-year overall survival and disease-free survival of the entire population were analyzed, and survival of the upstaging and the non-upstaging group was compared.

### 2.1. Study Design

The study was designed as a single-center and retrospective analysis of patients with early-stage NSCLC undergoing robotic lobectomy between January 2016 and December 2020. Data for the analysis were retrieved from our lobectomy database. According to the general inclusion criteria for this study, we enrolled patients diagnosed with clinical (T1–3, N0, M0) NSCLC who underwent lobectomy and radical hilum–mediastinal lymphadenectomy. The completeness of lymphadenectomy was evaluated in accordance with the IASLC definition regarding complete lymph node dissection of both N1 and N2 stations [9]. Patients who underwent surgery in 2021 were excluded to avoid the selection of patients with too short a follow up. Patients with tumors at clinical stages III–IV and who were subjected to sublobar resections and wedge resections were excluded. Patients who died within 90 days from the surgery were excluded. Patients who underwent preoperative chemotherapy or radiotherapy were excluded.

### 2.2. Preoperative Staging

Preoperative investigations included brain, thoracic, and abdominal computed tomography (CT) and F18-fluorodeoxyglucose positron emission tomography (FDG-PET) to establish the absence of multiple pulmonary lesions as well as of hepatic, adrenal, or brain metastases and to evaluate hilar and mediastinal lymph node status. When indicated, a brain MRI was performed to exclude brain metastasis. Hilar and mediastinal lymph nodes larger than 1 cm along the shortest axis or PET–CT avid, demonstrating a standardize uptake value >1.5, underwent endoscopic or endobronchial ultrasonography fine-needle biopsy to exclude metastatic involvement. Bone scintigraphy was performed if clinically indicated. Before surgery, all patients signed an informed consent to lobectomy.

### 2.3. Surgical Technique

All surgeries were performed using the Si da Vinci robot and the Xi da Vinci robot (Intuitive Surgical, Sunnyville, CA, USA). Patients were placed in the lateral decubitus position using single-lung ventilation with the hips fixed at the level of the table break and flexed to achieve maximum separation of the intercostal spaces. The Si da Vinci robot was positioned at the head of the patient. The Xi da Vinci robot was positioned at the back of the patient. We always proceeded by performing a 3 cm utility incision at the 5th or 6th intercostal space anteriorly of the latissimus dorsi. The wound was usually protected with a soft tissue retractor. Then, we placed the other three operative ports under direct-view guidance, usually at the 8th or 9th intercostal space. We then started docking the robot (Figure 1). We always used a 30°stereoscopic robotic camera. Under direct view, the bed-assistant started to introduce the operative robotics arms. Lobectomy was carried out with the usual technique. The pulmonary vein, pulmonary artery, and lobar bronchus were individually isolated and divided with a vascular three-line stapler. A parenchymal stapler was also used for the division of incomplete fissures. The lobe was retrieved with an endoscopic bag. In all patients, radical hilum–mediastinal lymphadenectomy was performed according to the guidelines. For right-side tumors, we systematically explored the paratracheal stations (2R and 4R), subcarinal stations (7), para-esophageal station (8), and inferior pulmonary ligament station (9). On the other side, lymphadenectomy of the aorto-pulmonary window (5–6), subcarinal stations (7), para-esophageal station (8), and inferior pulmonary ligament station (9) was usually carried out. At the end of the procedure, we usually inserted one chest tube using the camera port.

### 2.4. Follow-Up

We evaluated major morbidities until 30 days after surgery. One chest X-ray was performed one day after surgery and another one before removing the chest drain. The first outpatient follow-up was performed by thoracic surgeons after one month from the operation. The standard follow-up consisted of chest X-ray, laboratory testing including the measurement of tumor markers, and clinical examination. Further follow-up was managed by our oncological department. Patients underwent radiological, contrast-enhanced CT, and clinical examination every 6 months for the three years and then clinical and radiological evaluation annually. Adjuvant chemo- or radiotherapy was administered according to up-to-date guidelines [10].

### 2.5. Statistical Analysis

Statistical analysis was achieved with the assistance of an experienced biostatistician. The analysis was performed with SPSS 20 (IBM SPSS Statistics, IBM Corporation, Chicago, IL, USA). Descriptive statistics were calculated and are expressed as mean and standard deviation. Groups were compared using the t test for continuous variables and the chi-square test for categorical data. Survival analysis curves were achieved according to the Kaplan–Meier method. Disease-free survival and overall survival were compared using a log rank statistic.

## 3. Results

A total of 299 patients who underwent robotic pulmonary lobectomy were included in this retrospective study. Demographic and clinical features are reported in Table 1. Their median age was 67.4 ± 8.4 years. Comprehensively, 154 patients were male, and 145 were female. Former or active smokers were 223 (74.6%). Preoperative nodal disease was excluded in all the enrolled patients (cN0). The most common clinical stage was stage I, diagnosed in 259 of the 299 (86.6%) patients, while stage II tumors were identified in 40 (13.4%) of the 299 patients. At the time of surgery, all the patients underwent robotic lobectomy, with a prevalence of right and left upper lobectomy. The histopathological analysis results showed that the most frequent histological type was adenocarcinoma. Pathologic T distribution, number of dissected lymph nodes and nodal station, and their ratio are presented in Table 1.

### 3.1. Comparison between Upstaging and Non-Upstaging Groups

Of the 299 patients, pathologic absence of hilar or mediastinal lymph nodes (pN0) was confirmed in 244 patients (non-upstaging group), while histological analysis revealed hilar or mediastinal lymph nodes metastases in 55 patients (upstaging group). Therefore, the non-upstaging group included 81.6% of the patients, while the upstaging group comprised 18.4% of them. In the upstaging group, hilar metastatic lymph nodes without mediastinal lymph nodes involvement (pN1) were found in 29 patients (9.7%), and mediastinal metastatic lymph nodes, regardless of hilar nodal involvement (pN2), were found in 26 patients (8.7%). The distribution of the dissected and metastatic lymph nodes and nodal stations in the upstaging group is presented in Table 2.

The non-upstaging and upstaging groups were further compared. A comparison of the demographical and surgical features between the two groups is presented in Table 3. The two groups demonstrated to be homogeneous for age at presentation (*p* = 0.76), gender (*p* = 0.16), smoking habits (*p* = 0.59), clinical T stage at presentation (*p* = 0.1) as well as surgical side (0.47), type of lobectomy (0.44), and histology (0.76). In the upstaging group, the absolute number of dissected lymph nodes demonstrated to be larger than in the non-upstaging group (*p* < 0.001). Conversely, the number of dissected nodal stations demonstrated to be homogeneous in the two populations (*p* = 0.2). The dissected lymph nodes-to-nodal station ratio demonstrated to be larger in the non-upstaging group (*p* = 0.003).

### 3.2. Survival Analysis

Follow-up was continued for a median of 27.6 months (range, 1 to 67 months). Recurrences were found during follow-up in 94 patients (31.4%), and 40 patients (13.4%) died. No significant differences were found in follow-up duration between the non-upstaging and the upstaging group (*p* = 0.52). In the 244 patients of the non-upstaging group, recurrence was found during follow-up in 62 patients (25.4%), and 26 patients (10.7%) died. In the 55 patients of the upstaging group, recurrence was found during follow-up in 32 patients (58.2%), and 14 patients (25.5%) died.

In the total population, the 3-year overall survival was 82.8% (Figure 2a), while the overall survival stratified for the different pathological stages is presented in Figure 2b. The 3-year overall survival in the non-upstaging group was 86.2%, while in the upstaging group, it was 71.4%. As shown in Figure 3, the difference in overall survival between the non-upstaging and the upstaging group was statistically significant (*p* = 0.004).

In the total population, the 3-year disease-free survival was 61.3% (Figure 4). The 3-year disease-free survival in the non-upstaging group was 68.7%. The 3-year disease-free survival in the upstaging group was 34.9%. As shown in Figure 5, the difference in disease-free survival between the non-upstaging and the upstaging groups was statistically significant (*p* < 0.0001).

## 4. Discussion

Although preoperative staging techniques have been strongly improved, the number of hidden positive lymph nodes is still high [11]. Therefore, a correct lymphadenectomy should be one of the main goals during lung cancer surgery [12]. Lobectomy followed by mediastinal lymphadenectomy is the standard treatment in early-stage non-small cell lung cancer [13]. Until two decades ago, the standard surgical approach to perform anatomical lung resections was thoracotomy, but in the last few years, the use of minimally invasive techniques has progressively increased [14]. First with video-assisted thoracic surgery (VATS) and, more recently, using robot-assisted thoracic surgery (RATS), minimally invasive techniques have become more frequent in routine surgical practice, ensuring lesser morbidity and shorter hospital stays compared to thoracotomy [15]. The robotic approach represents a technological evolution of the VATS procedure. It has some technical advantages such as a better view of the operative field, a simpler use of the instruments, and the execution of more precise movements and of a larger range of movements thanks to the wide angle of maneuverability of the instruments, which is even superior to that of the human hand [16].

One of the first experiences with robotic thoracic surgery was described by Melfi et al., with good perioperative results in 12 patients [17]. In the following 20 years, robotic surgery has rapidly expanded into the surgical practice of thoracic surgeons, and the last results showed that the robotic approach has better perioperative outcomes compared to thoracotomy [18].

In this retrospective analysis of a single-center robot-assisted thoracic surgical experience, a total of 559 robot-assisted lobectomies performed from May 2016 to December 2021 were examined to evaluate the oncological results of robot-assisted lobectomies with radical lymphadenectomy in the treatment of early-stage lung cancer. Out of the 559 RATS lobectomies, radical lymphadenectomy was achieved in 422 cases, but the selection criteria for this study were met only by 299 of the 559 patients. Each of the 299 patients enrolled in the study underwent lobectomy followed by radical lymphadenectomy for early-stage NSCLC. In these patients, extensive pre-operative exclusion of hilar or mediastinal nodal metastases was accomplished through contrast-enhanced CT, PET–CT, and in suspected cases, through endobronchial or endoscopic ultrasonographic fine-needle biopsy, as recommended in the most recent guidelines. Nonetheless, 55 patients (18.4%) demonstrated to have hilar or mediastinal metastases at histopathological examination after surgery. The rate of pN1 was 9.7%, while the rate of pN2 disease was 8.7%. These upstaging rates were similar to the one described in previous reports examining lymphadenectomy during RATS or thoracotomy. Particularly, in a recent analysis by Zirafa et al., upstaging results in NSCLC patients treated by thoracotomy or RATS were compared [19]. In this study, nodal upstaging was found in 41 out of 212 (19.3%) patients. In another report by Toosi et al., nodal upstaging was demonstrated in 41 (14.3%) of the 287 patients who underwent robotic lobectomies [20].

The quality of lymphadenectomy has a crucial role in the oncological outcomes of NSCLC patients [21]. An adequate hilar and mediastinal nodal dissection is necessary to prevent downstaging, lack of adjuvant treatment, and undertreatment [22]. Among the most relevant indicators of lymphadenectomy quality are the number of dissected hilar and mediastinal nodal stations and the number of harvested nodes [23]. In our series, the mean number of dissected nodal station was 4.4 ± 1.3, with a mean number of dissected lymph nodes of 13.5 ± 7.7. These results are consistent with previous reports. Particularly, in this report, the mean number of harvested lymph nodes as well as the dissected lymph nodes-to-nodal stations ratio were larger in the upstaging group than in the non-upstaging group. This result is consistent with previous findings [24,25]. Several hypotheses may explain this result. In patients with nodal involvement, surgeon attention may be devoted to harvest the greatest number of hilar and mediastinal lymph nodes, as this procedure is expected to improve the patients’ therapeutic chances, ameliorating their prognosis. Although this is an intriguing explanation, it may not fit our results, as patients were all deemed as N0 before surgery, and radical lymphadenectomy was offered to all patients. Nonetheless, the number of nodal stations dissected was homogeneous in the two groups. Conversely, the larger number of lymph nodes in the upstaging patients may be the result of the metastatic process following the release of a complex array of chemokines or lymphangiogenic factors by the tumoral tissue, which were demonstrated to have a prognostic impact [26]. As the biologic reasons of lymph node number variability is unknown, the prognostic impact of the number of resected lymph nodes is still controversial [27,28]. According to the current recommendations at least 10 lymph nodes must be dissected in order to obtain a proper staging [29,30].

However, in our analysis of the upstaging group, 25 out of 55 patients (45%) presented only 1 metastatic node out of an average of 15.8 dissected lymph nodes. Thus, dissecting a lower number of nodes may lead to an unacceptable risk of downstaging and undertreatment.

At our institution, the robotic thoracic surgery program began in 2016. For this reason, the follow-up period in this study ranged from 1 to 67 months, with a mean follow-up duration of 27.6 months. The 3-year overall and disease-free survival was 82.8 and 61.3, respectively. Generally, the 3-year overall survival in early-stage lung cancer after robotic lobectomy and radical lymphadenectomy ranges from 85 to 90% [31]. In our series, the overall survival appeared to be quite short. This finding may be explained by the histological characteristics of the tumors in our population. Among the 299 patients, 27 patients demonstrated a pathological T stage ≥3. Nonetheless, the nodal upstaging rate was one of the highest, revealing that 55 out 299 patients had locally advanced diseases. As demonstrated in Figure 4, the 3-year overall survival stratified on a tumor-stage basis showed a 3-year overall survival of 90% for stage I tumors. These results are consistent with the ones described in previous reports [32]. Both overall and disease-free survival demonstrated to be statistically different when comparing the upstaging and the non-upstaging group. Therefore, nodal upstaging has a significant prognostic impact. Basically, this retrospective study regarding the oncological outcomes of robotic surgery at our department demonstrated similar results to the ones previously described [33].

This study has several limitations, including the retrospective nature of the analysis and the relatively short time of the follow-up. At our institution, the robotic program started in 2016, thus a 5-year follow-up was completed only for a limited number of enrolled patients. To reduce the impact of the short follow-up on the survival analysis, both overall and disease-free survival periods were analyzed in the three post-operative years.

In conclusion, robotic lobectomy demonstrated to be safe and feasible for NSCLC patients. The oncological outcomes we described are consistent with those in previous literature, with a 14.3% nodal upstaging rate. For the upstaging group, the dissected lymph node number and the dissected hilar and mediastinal station ratios were significantly higher than those determined for the non-upstaging group. The upstaging group demonstrated significantly lower 3-year overall and disease-free survival. Thus, radical lymphadenectomy aimed at removing the highest number of lymph nodes should be pursued.

## Figures and Tables

**Figure 1 jcm-11-02173-f001:**
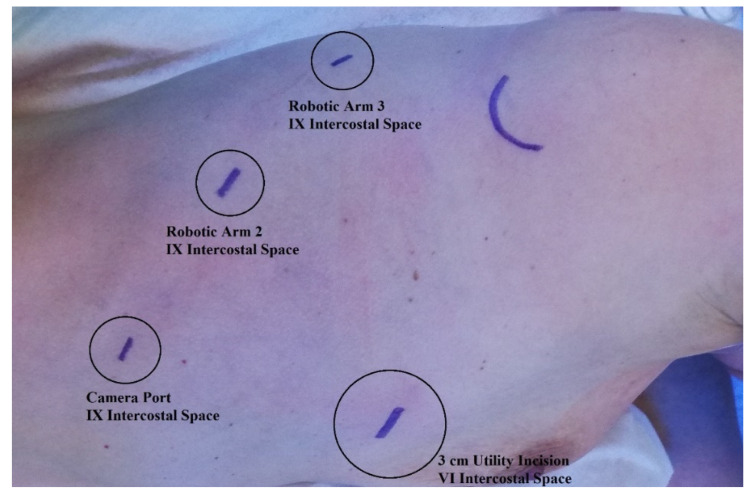
Robotic ports placement.

**Figure 2 jcm-11-02173-f002:**
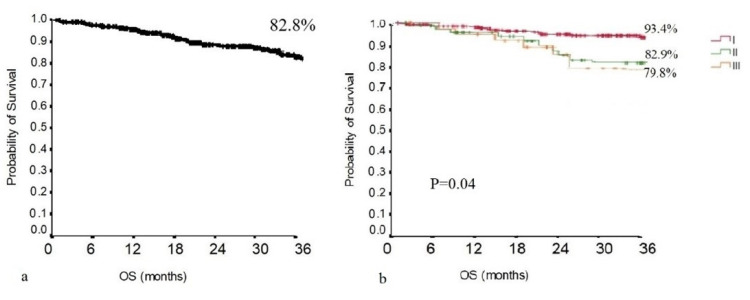
(**a**) OS of the population; (**b**) OS of the population stratified for disease stage.

**Figure 3 jcm-11-02173-f003:**
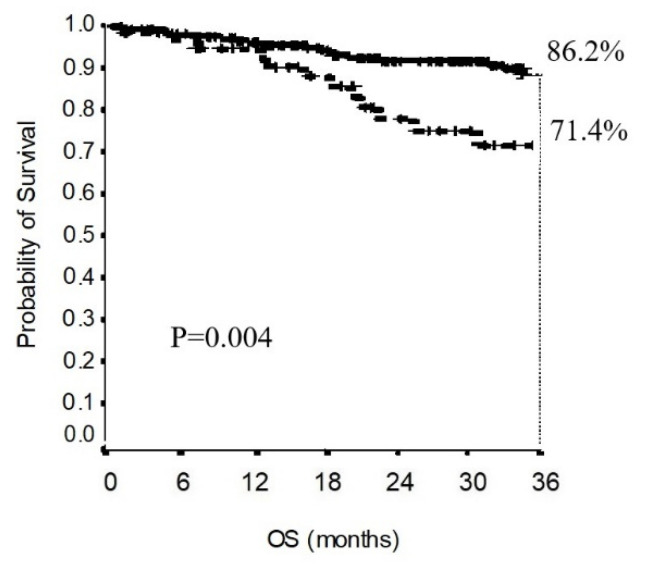
Comparison of the OS of the upstaging group with that of the non-upstaging group.

**Figure 4 jcm-11-02173-f004:**
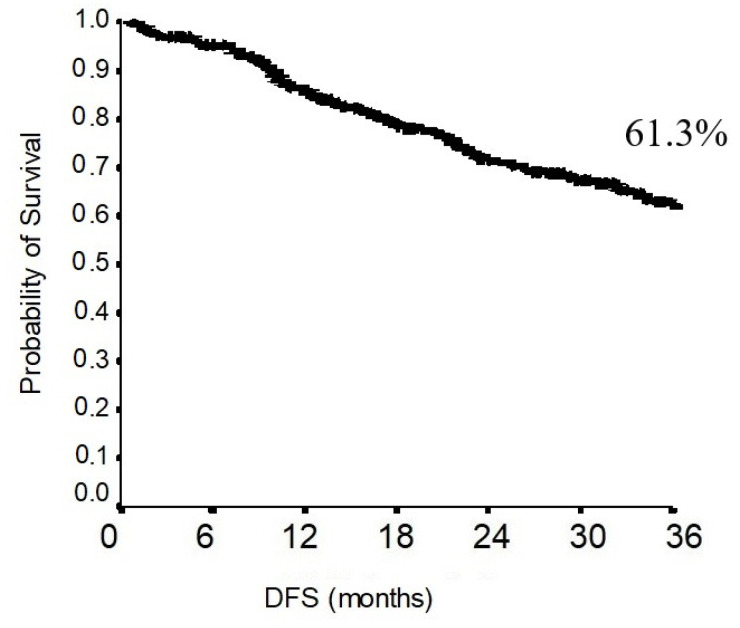
DFS of the population.

**Figure 5 jcm-11-02173-f005:**
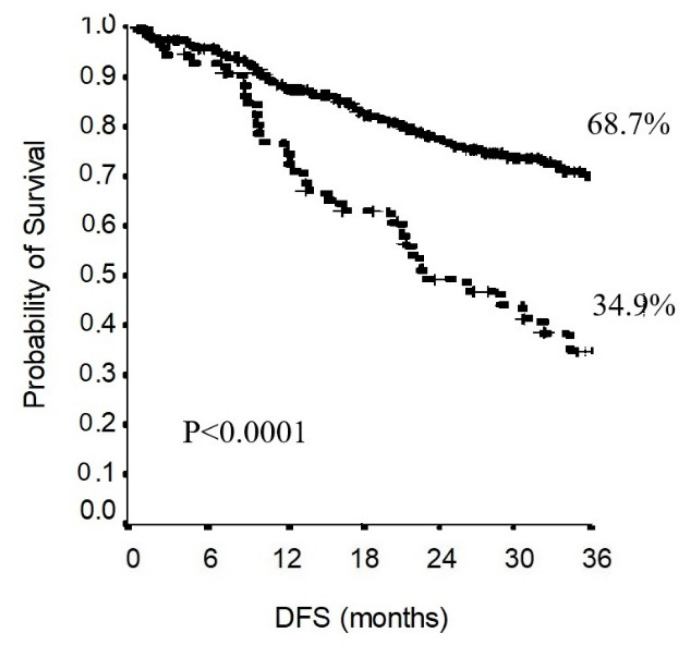
Comparison of the DFS of the upstaging group with that of the non-upstaging group.

**Table 1 jcm-11-02173-t001:** Demographic and surgical features of the patient population.

	All Patients (*n* = 299)
**Age (years)**	67.4 ± 8.4
**Gender (M/F)**	154/145
**Active or former smoker (*n*, %)**	221 (73.9)
**Preoperative T (*n*, %)**	
**I**	149 (50.2)
**II**	123 (40.8)
**III**	27 (6.3)
**Side (R/L)**	160/139
**Lobectomy (*n*, %)**	
**RUL**	92 (30.8)
**ML**	16 (5.3)
**RLL**	52 (17.4)
**LUL**	86 (28.8)
**LLL**	53 (17.7)
**Dissected lymph node (mean ± SD)**	13.5 ± 7.7
**Dissected nodal stations (mean ± SD)**	4.4 ± 1.3
**Dissected lymph node/nodal stations (mean ± SD)**	3.1 ± 1.7
**In-hospital stay (days)**	6.2 ± 2.3
**Histology (*n*, %)**	
**Adenocarcinoma**	262 (87.6)
**Squamous Cell Carcinoma**	37 (12.4)
**Pathologic T (*n*, %)**	
**I**	133 (44.5)
**II**	139 (46.5)
**III**	21 (7)
**IV**	6 (2)
**Follow-up duration (months)**	27.6 ± 16.5

**Table 2 jcm-11-02173-t002:** Mediastinal and hilar nodal upstaging rate of the population.

	Nodal Upstaging (*n* = 55)	Hilar Upstaging (*n* = 29)	Mediastinal Upstaging (*n* = 26)	*p*-Value
**N of resected lymph node**	17.2 ± 8.5	16.1 ± 9.9	18.3 ± 6.7	0.36
**N of resected nodal station**	4.7 ± 1.2	4.4 ± 1.2	5.0 ± 1.2	0.08
**N of metastatic lymph node**	2.3 ± 2.1	1.4 ± 0.7	3.3 ± 2.6	0.001
**N of metastatic nodal stations**	1.5 ± 0.8	1 ± 0.2	1.9 ± 1.0	<0.001

**Table 3 jcm-11-02173-t003:** Comparison of the demographic and surgical features of the upstaging and non-upstaging groups.

	Upstaging Group (*n* = 55)	Non-Upstaging Group (*n* = 244)	*p*-Value
**Age (years)**	67.7 ± 7.5	68.0 ± 8.3	0.76
**Gender (M/F)**	33/22	119/121	0.16
**Active or former smoker (*n*, %)**	42 (76.4)	179 (73.4)	0.59
**Preoperative T (*n*, %)**			0.10
**I**	20 (36.4)	129 (52.9)	
**II**	29 (52.7)	94 (38.5)	
**III**	6 (10.9)	21 (8.6)	
**Side (R/L)**	27/28	133/111	0.47
**Lobectomy (*n*, %)**			0.44
**RUL**	13	79	
**ML**	5	11	
**RLL**	9	43	
**LUL**	19	67	
**LLL**	9	44	
**Dissected lymph node (mean ± SD)**	17.2 ± 8.5	12.7 ± 7.3	<0.0001
**Dissected nodal stations (mean ± SD)**	4.7 ± 1.2	4.3 ± 1.3	0.2
**Dissected lymph nodes/nodal stations (mean ± SD)**	3.7 ± 1.6	2.9 ± 1.7	0.003
**In-hospital stay (days)**	6.0 ± 1.8	6.2 ± 2.5	0.76
**Histology (*n*, %)**			0.34
**Adenocarcinoma**	46 (83.6)	216 (88.5)	
**Squamous Cell Carcinoma**	9 (16.4)	28 (11.5)	
**Pathologic T (*n*, %)**			
**I**	18 (32.7)	115 (47.1)	
**II**	31 (56.4)	108 (44.3)	
**III**	5 (9.1)	16 (6.6)	
**IV**	1 (1.8)	5 (2)	
**Follow-up duration (months)**	26.3 ± 15.3	27.9 ± 16.8	0.52

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
