# Peer review of "Oncological Outcomes of Robotic Lobectomy and Radical Lymphadenectomy for Early-Stage Non-Small Cell Lung Cancer"

_jcm, 2022, doi:10.3390/jcm11082173_

Round 1

Reviewer 1 Report

Overall, the authors' described an insightful study, but several changes should be considered before publication is finalized.  I encourage the author to see the attachment for specific revisions, to further enhance readability and highlight academic points. 

Abstract
The authors’ abstract communicated the background, methods, results and conclusion in a systematic fashion. The
results outlined the outcome of OS and DFS between upstaged and non upstaged cohorts , but the statistical values
were not provided to highlight the statistical difference. This should be added in the results and reinforced in the
conclusion as well. Grammar construct needs attention.

Introduction

The authors’ provided a brief overview of the current approaches for anatomical lobectomies, while highlighting the
advantages of the minimal invasive approach to lobectomies, in comparison to the open approach. The inherent
benefits of robotic surgery were described, but current literature on lesser blood loss and improved lymph node
harvest in the robotic arm versus the minimally invasive arm, should be referenced.1-2 Merrit et al also describe the
benefits of the robotic approach in comparison to open, after comparing lobectomy outcomes in high risk patients.3
The authors should consider adding these references. Grammar construct in the introduction should be addressed.

Methods

The author described a methodical approach to the study and highlighted the retrospective aspects of this study. The
study’s focus was concisely described, thus the authors should consider moving this paragraph (line 58-69) to the
end of the introduction.

Secondly, the authors’ study design mentioned “patients diagnosed with clinical N0 NSCLC at stage I-II” in Line 74.
Please consider substituting for “patients diagnosed with clinical (T1-3, N0, M0).” Moreover, T3 disease requires
brain imaging, were these patients included, the authors should specify in their inclusion/exclusion criteria

Minor changes include addressing spelling errors (Line 59 (December is spelled incorrectly and should spelled out)
and specify company and location (Intuitive Surgical, Sunnyville, CA) in Line 93

Consider creating a figure illustration, displaying your port sites.

Results:

The results were well articulated. I suggested a few changes to improve data display.

Line 136-137: All patients after pre-operative 136 staging were in stage I or II according to the 8th TNM
classification; This is already mentioned in the methods, and does not need to be repeated. Follow up duration
should specify time metric (27 months) in the table. Provide p values for Figure 1a and b

Conclusion

The conclusion clearly summarized the study’s findings and incorporated references to support the validity of the
study. I suggest the author omit hilum from line 198 and cite Ginsberg’s paper on randomized trial between
lobectomy vs limited resection for early stage lung cancer.4

To date, there is not a study that confirms the superiority of the robotic approach to the open or VATs approach.
Robotic approaches improve lymph node harvest, which contribute to upstaging as described in your paper and
referenced from
Frontier. It is not proven to be “better” especially if you consider the cost burden to conduct these
operations.

Line 254-255 :At least 10 lymph nodes are needed to stage adequately.5

275-279:The authors should consider elaborating on the study’s limitation, with focus on the limitations of clinical
staging. I suggest elaborating on the sensitivity, specificities of radiographic imaging for lung cancer work up. This
further highlights the advantages of robotic approach.

Consider adding prospective future study ideas; upstaging in robotic approach vs video-assisted? Cost benefit
difference?

1. (Ma, J., Li, X., Zhao, S.
et al. Robot-assisted thoracic surgery versus video-assisted thoracic surgery for lung
lobectomy or segmentectomy in patients with non-small cell lung cancer: a meta-analysis.
BMC
Cancer 21, 498 (2021).
https://doi.org/10.1186/s12885-021-08241-5)
2. Su Yang, Wei Guo, Xingshi Chen, Han Wu, Hecheng Li, Early outcomes of robotic versus uniportal video-
assisted thoracic surgery for lung cancer: a propensity score-matched study,
European Journal of Cardio-
Thoracic Surgery, Volume 53, Issue 2, February 2018, Pages 34835

3. Kneuertz, P.J., D’Souza, D.M., Moffatt-Bruce, S.D.
et al. Robotic lobectomy has the greatest benefit in
patients with marginal pulmonary function.
J Cardiothorac Surg 13, 56 (2018).

https://doi.org/10.1186/s13019-018-0748-z

4. Ginsberg RJ, Rubinstein LV. Randomized trial of lobectomy versus limited resection for T1 N0 non-small cell
lung cancer. Lung Cancer Study Group. Ann Thorac Surg. 1995 Sep;60(3):615-22; discussion 622-3. doi:
10.1016/0003-4975(95)00537-u. PMID: 7677489.

5. Samayoa AX, Pezzi TA, Pezzi CM, Greer Gay E, Asai M, Kulkarni N, Carp N, Chun SG, Putnam JB Jr.
Rationale for a Minimum Number of Lymph Nodes Removed with Non-Small Cell Lung Cancer Resection:
Correlating the Number of Nodes Removed with Survival in 98,970 Patients. Ann Surg Oncol. 2016
Dec;23(Suppl 5):1005-1011. doi: 10.1245/s10434-016-5509-4. Epub 2016 Aug 16. PMID: 27531307.

Author Response

We thank the reviewer for the constructive and useful comments. We responded point by point below.

  • The abstract was incremented with the statistical values in order to highlight the results.
  • The introduction section was reviewed and we added some sentences that described the papers suggested by the reviewer.
  • As suggested by the reviewer we changed the sentence “patients diagnosed with clinical N0 NSCLC at stage I-II” in Line 74 with “patients diagnosed with clinical (T1-3, N0, M0).” We are agree that it is more comprehensive. We included in the study also patients with cT3 disease. All the patient included in the study underwent brain images (contrast enhanced CT scans or MRI) to exclude brain metastasis. We better specified in the method section. We revised the spelling errors and we added the robotic company information. We added a figure with the robotic ports placement.
  • P value for figure 1b and the follow up duration in the table were added.
  • As suggested by the reviewer we added information regarding the limitations of the clinical preoperative staging in the introduction. Indeed, despite the preoperative staging guideline are well established, the nodal upstaging rate remains high, as confirmed by the literature. We are aware that there are not in the literature studies that confirm the superiority of the robotic surgery compared to the open or VATS, we reviewed the conclusion according to this view. Despite it is not confirmed by the literature, we suppose that the robotic surgery allows a better postoperative staging with more accurate postoperative therapy indication and better oncological outcomes.
  • Line 254-255 “At least 10 lymph nodes are needed to stage adequately” was changed adding the reference suggested.
  • We thank the reviewer for the future study ideas that we surely will take into consideration.

Reviewer 2 Report

Dear authors,

 you spend a lot of effort in preparation of this manuscript and are according to the STROBE Checklist.

The experiences in treatment of NSCLC via RATS are very interesting and the presentation is clear. Grammar and syntax are fine and the manuscript is easily readable.

The number of more than 500 patients, which were treated by RATS is impressive and the number of 299 patients which met the inclusion criteria of the study is still imposing.

I got few annotations:

-           In the abstract thoracotomy is mentioned to be approach which is gold standard for lobectomy.            This      might be controverse, 20 years after implementing VATS as an approach.

-           In the abstract you state that there are differences in OS and DFS between the groups but they do            not mention which group is superior.

-           In the introduction you present the 5-year-survival of patient with NSCLC for all stages. But the      cohort got NSCLC in early state. The authors should give the 5-year survival of patients with early      stage either.

-           You state controversities regarding oncological radicality of VATS compared with thoracotomy. Actually, lots of studies support al least equality of approaches.

-           The inclusion and exclusion criteria enable good comparability with your groups. As well as comparability with other cohorts.

-           Was FDG-PET the staging for absence of brain metastases? Have the authors performed MRT?

-           What was the cN-status after PET-CT and before “exclusion” of positive nodal status via EBUS-TBNA?

-           Was the mortality after 30 and 90-days captured?

-           You state RATS to be a safe approach for lobectomy. What was the morbidity in your group?

-           In Table 1 the measure of follow up is missing.

-           The results are clearly presented. And the discussion is

-           The figures of survival are very well and give a good illustration.

Your study seem to be a great prove to elevate the mediastinal staging up to mediastinoscopy after bronchoscopy without positive findings.

Author Response

We thank the reviewer for the useful comments and below we responded point by point. Overall, we decided to focus in this paper on oncological outcomes do not report the perioperative outcomes. As suggested by the reviewers we better specified the methods of this study. Then, we are honored by the consideration given to us by the reviewer regarding our robotic activity.

  • We reviewed the sentence and we rewrote it in a more comprehensive way.
  • We reviewed the sentence and we added the p-values.
  • The estimated 5 year OS of early stage was added in the introduction.
  • We decided to use the terms “controversies” because in the literature there are very few studies that compared the long term outcomes of these two techniques of which almost none prospective and randomized. As stated by the reviewer, we are aware that various retrospective analyses showed comparable outcomes between open and VATS.
  • We better specified in the methods section the radiological preoperative staging, including the use of MRI to exclude brain metastasis when indicated.
  • According to the last European guidelines, we used EUS-FNA and EBUS-TBNA to exclude nodal metastasis. We selected only patients that reported after the preoperative staging (radiological and endoscopic) the cN0.
  • In this study we decided to focusing on the oncological outcomes and to prevent some bias of selection we exclude patient that die after 30 or 90 days.
  • In this study, we decided to report only oncological outcomes and results in terms of nodal dissection. In a previous paper where we compared the perioperative outcome of robotic surgery with VATS and thoracotomy, we reported a postoperative major morbidity rate until 30 days after surgery that was significantly higher in thoracotomy group compared to the VATS group and the robotic group (23, 14,5% Vs 6, 5,4% Vs 13, 5,5%; p=0.04). (Gallina FT, Melis E, Forcella D, et al. Nodal Upstaging Evaluation After Robotic-Assisted Lobectomy for Early-Stage Non-small Cell Lung Cancer Compared to Video-Assisted Thoracic Surgery and Thoracotomy: A Retrospective Single Center Analysis. Front Surg. 2021;8:666158. doi:10.3389/fsurg.2021.666158)
  • The measure of follow up was added.